# Cross-Border Dialogues: A Collaborative Instructional Design Inquiry to Promote Equity and Diversity

Zheng Zhang [1,*], Icy Lee [2], Helen Wan Yu Chan [2], Qi Guo [1], Angela Kuan [2], Jessica Sum Laam Lee [2], Qianhui Ma [1], Natalie Ching Tung Ng [2] and Rozan Trad [1]

[1] Faculty of Education, Western University, London, ON N6A 1G7, Canada; qguo88@uwo.ca (Q.G.); qma63@uwo.ca (Q.M.); rtrad@uwo.ca (R.T.)
[2] Faculty of Education, Chinese University of Hong Kong, Shatin, NT, Hong Kong, China; icylee@cuhk.edu.hk (I.L.); wchan3200@163.com (H.W.Y.C.); angelakuan313@gmail.com (A.K.); jessicalee0801214@gmail.com (J.S.L.L.); ngchingtung.com@gmail.com (N.C.T.N.)
* Correspondence: zzhan58@uwo.ca

**Abstract:** The COVID-19 pandemic complicates ingrained educational inequalities around the globe and foregrounds the pertaining challenges that teachers have encountered due to school closures and the shift to distance learning. This cross-border teacher education project intended to examine how academics and pre-service teachers in different geographic locales could collaboratively explore equitable learning opportunities for diverse learners through the use of critical media literacies to respond to interconnected global crises. In this six-week cross-border teacher education project, we recruited four Mandarin and English literacy teacher candidates in Hong Kong to interact with one another and one Canadian professor as part of the teacher preparation phase of a larger-scale cross-border research project that connects youth from Hong Kong and Canada in a social networking space. For the purposes of teacher professional development, the Hong Kong teacher candidates and Canadian researchers engaged in collective exploration of how instructional designs in literacy education could promote equitable learning opportunities for diverse learners. Findings show that the cross-border teacher education project supported teacher candidates' development of pedagogical skills and espoused their agency in promoting educational equity and collective problem-solving through critical media literacies. Findings relate the teacher candidates' shifted perspectives from focusing on students' decontextualized language skills to nurturing critical media skills. Changing from a deficit-oriented view about what literacy learners could not do, the teacher candidates also adopted an asset-oriented view about the linguistic and cultural repertoires that diverse learners could bring to literacy classrooms.

**Keywords:** critical media literacy; teacher education; professional development





## 1. Introduction

The information age has shifted the focus of literacy education from teaching reading and writing in official, standardized, and printed formats to nurturing critical users of new media that have substantially changed most people's ways of living and communicating [1–3]. The COVID-19 pandemic foregrounds the global digital divide and the pertaining challenges that teachers have encountered due to school closures and the shift to online learning e.g., [4,5]. The global pandemic has also stressed the interconnectivity between people across geographic locales and the importance of collective problem solving for living well together e.g., [6]. This study responded to the scholarly calls to envision literacy education that has the capacity to critically address global issues and rapid technological changes and collectively explore new ways of enacting equity in and through education e.g., [7]. To pilot teacher professional development for a larger cross-border new media literacy project that connects Canadian and Hong Kong youth, this teacher education project connected one Canadian professor and four Hong Kong pre-service teachers via the virtual spaces of Teams

and Google Docs. This teacher education project intended to examine how academics and pre-service teachers in different geographic locales could collaboratively explore equitable learning opportunities for diverse learners by using critical media literacies to respond to interconnected global crises. This project asked: (1) How did Canadian researchers and Hong Kong pre-service literacy teachers collaboratively explore equitable learning opportunities for diverse learners using critical media literacies? (2) How did Hong Kong teacher candidates perceive the effects of cross-border, online professional learning?

**2. Literature Review**

We conducted an extensive literature review of empirical studies on teacher education and critical media literacy and identified three themes: teacher candidates' increased awareness of using critical media literacies through teacher education courses on critical media literacy, their willingness to engage in media analysis and incorporate critical media literacies in their classes, and the challenges for teacher candidates in applying critical media literacies in their courses.

Researchers pointed out that teacher education courses or conversations on critical media literacy could provide opportunities for pre-service teachers to raise their awareness of issues associated with gender, race, identity, media representations, advertising, and consumerism to challenge dominant ideologies and social inequity and injustice e.g., [8–13]. For instance, Flores-Koulish's [14] study examined pre-service teachers' knowledge about media culture and media literacy pedagogy and curriculum through a Madonna video, "Material Girl". The findings relate the depth of participants' video analyses and their ability to identify issues associated with gender stereotypes and media representation. The study also reported the pre-service teachers' "overall media literate deficits" (p. 246) and their fear of political engagement. Marlatt [12] designed an intervention project for pre-service teachers to critically analyze ideologies embedded in a published report by the U.S. Secretary of the Interior. Cross-checking report sources, pre-service teachers found that misleading language in media messages was one of the reasons for the spread of false information among the public. Findings also show that pre-service teachers positioned themselves against the mainstream media when the media messages failed to make information transparent and obscured "political motivations and financial interests" (p. 97). Share [13] created the *Through Others' Eyes* assignment for pre-service teachers to question media portrayals across a range of areas, such as body image, immigration, domestic violence, alcoholism, religion, stereotypes, and the intersections of racism, sexism, classism, and homophobia. The teacher candidates were also encouraged to reflect on others' postings about an aspect of their identities. The teacher candidates' critiques of negative representations helped them explore the influences of visual images and the deep connections between media, power, and identity.

Studies reported that pre-service teachers were positive about constructing critical media literacy lessons in their own classrooms [8,11,13,15]. According to Laughter [11], pre-service teachers would willingly implement critical media literacy in their own classrooms for the sake of social justice. In Robertson and Hughes's [15] study, pre-service teachers indicated that, as the "agents of change" (p. 44), they would support their students in exposing inequity and injustice in and outside school communities. In the survey, the pre-service teachers in Robertson and Hughes's study reported that it was very important to help their future students become critically media literate citizens. However, the study found that the teacher candidates focused on one aspect of media messages without making "connections across issues" (p. 49). For example, the pre-service teachers captured only one hidden message of using topless male models in clothing advertisements; however, they did not identify other issues, such as body perfection encoded in media as a mainstream value and its impact on people.

Studies show that teacher candidates' perceptions of critical media literacy and its use might hinder their incorporation of critical media literacy in real classrooms e.g., [10,13–17]. In Share's [13] study, some pre-service teachers found it challenging to probe deeply into

racism- and privilege-related issues. Similarly, Kellner and Share [10] also found that some pre-service teachers found it challenging to acknowledge the role of power and privilege in "social systems of oppression" (p. 67). The pre-service teachers in Amory's [16] study appreciated the significance of using interactive games to construct knowledge but were concerned about not fully understanding how to use video games to mediate learning processes. Teacher participants in Flores-Koulish's [14] study were able to identify the basic issues related to gender stereotypes while also missing the mark on internalizing these concepts. In Robertson and Hughes's [15] study, only one pre-service teacher elaborated on her understanding of critical media literacy; in contrast, her peers indicated that they had not previously paid attention to certain sociopolitical issues, such as media construction.

Limited resources from schools and standardized ways of assessing and teaching students would also constrain pre-service teachers' efforts to design and teach critical media literacy lessons in their own classrooms e.g., [10,14,15,17]. For example, in Robertson and Hughes's [15] study, pre-service teachers reported that digital media devices (e.g., projectors and Smart Boards) are not accessible in every classroom. Share et al.'s [17] paper addressed the lack of support for critical media literacy education in school departments and administration. Flores-Koulish [14] was concerned that the "accountability movement" (p. 239) might constrain pre-service teachers' efforts and agency to engage in critical media literacy education. Flores-Koulish noted that under this accountability movement, students were judged by scores on standardized tests and teaching practices focused more on test preparation and content delivery. Sharing Flores-Koulish's concern with accountability and standardization, Kellner and Share [10] also contended that homogenizing "what is to be taught" (p. 79) would limit expansive conceptualizations of literacy and teachers' integration of critical media literacy in classrooms.

To conclude, existent literature shows that pre-service teachers used digitally mediated tools for their professional learning and created digital messages, postings, stories, and images to reflect on their critical media literacies. However, Fajardo's [18] review of critical literacy studies shows that teachers' professional learning programs have not encouraged teachers to address socio-political issues in their language classrooms. Pre-service teachers need to be critical media literate themselves before they could deepen their students' criticality about power, privilege, and ideology [13]. Responding to this scholarly call, our cross-border teacher education project also addressed the scarcity of literature on using critical media literacies to respond to interconnected global crises. The study was hence designed as a collaborative professional inquiry that tried to close the gaps between academic research and teachers' classroom practices to attend to "the digital landscape and global shifts of the twenty-first century" [19], p. 12.

## 3. Theoretical Underpinning

The cross-border teacher education project is theoretically undergirded by critical media literacies. Instructional designs involve the creation of learning experiences, environments, and materials that promote learners' acquisition and application of knowledge and skills [20]. Forms of instructional designs include the creation of instructional materials, modules, and lessons. Instructional designs are also explicit in their choice and use of procedures, methods, and devices to bring about productive learning [21]. In this study, we concurred that critically oriented literacy teaching should be a participatory and collaborative project [19]. The shared leadership in instructional design included both academics and teacher candidates to collectively brainstorm and select topics, materials, and activities.

Our cross-border literacy teacher education aimed to explore with teacher candidates how critical media literacies could be used to promote equity in increasingly diverse educational contexts. Critical media literacies expand the notion of literacy as traditional print forms and include "different forms of mass communication, popular culture, and new technologies" [9], p. 111; hence, the use of its plural form in this paper.

There are five concepts that align with media literacy, according to the Center for Media Literacy of America e.g., [11,22] (1) the principle of non-transparency—all messages

are constructed; (2) codes and conventions—media messages are constructed using a creative language with its own rules; (3) audience decoding—different people experience the same media message differently; (4) content and message—media have embedded values and points of view; and (5) motivation—media are organized to gain profit and/or power. Hence, a critical media literacies approach expects students to develop a critical understanding of how corporate for-profit media are driven by their political and economic interests [3]. Students can also be empowered to create "alternative, non-profit media" [3], p. 261, that problematize media texts and narratives [8].

Critical media literacies aim to develop the public's skills to promote civil participation and democratization [22]. Torres and Mercado [3] specified that critical media literacies are "founded on the legitimate role of media to serve the public's right to be truly informed, and thereby serve democracy" (p. 260). Critical media literacies focus on teaching "analysis of relationships between media and audiences, information and power" [8], p. 51; for example, demystifying stereotypes and assumptions underlying media texts. Critical media literacies could cultivate students' skills in analyzing media codes and conventions, their abilities to demystify stereotypes, assumptions, dominant values, and ideologies, and capabilities to interpret the multiple meanings and messages underlying media texts [22]. Critical media literacies are not only concerned with students' critical media analysis but also help "create good citizens" who are well-motivated and competent to participate in social life [22], p. 372. As Joanou [23] stated, critical media literacies support students' critical engagement with the media and popular culture and their connection between critical theory and media texts to "transform their classrooms into sites for social change" (p. 41).

An important overarching goal for pre-service teachers is for them to take responsibilities to help students become "actively engaged in alternative media use and development" [3], p. 261. Critical media literacies could prepare pre-service teachers to better support "their students in critical inquiry *with* and *about* information communication technologies (ICTs) and popular culture" [1], p. 319. Critical media literacies could also help educators "rethink teaching and learning as political acts of consciousness raising and empowerment" [1], p. 319. Therefore, it could offer in-service and pre-service educators pedagogical strategies to "strengthen civic engagement and reassert the promise of democracy with an informed and empowered citizenry" [1], p. 319.

Funk et al. [1] proposed a critical media literacy framework to support pre-service teachers' learning *with* and *about* ever-evolving media and technologies. The framework includes six aspects of critical media literacy: (1) All media texts are socially constructed by individuals or groups within social contexts; (2) all media have their own languages, with specific grammar and semiotics; (3) individuals and groups understand media texts similarly and/or differently depending on multiple contextual factors; (4) media and media messages support or disrupt dominant hierarchies of power, privilege, and pleasure; (5) all media texts serve a purpose that is shaped by the creators and/or systems within which they operate (e.g., commercial or government); and (6) media culture concerns social and environmental justice and perpetuates or challenges positive and/or negative ideas about people, groups, and issues.

The critical media literacy framework could support pre-service teachers' critical interrogation of the role of media, power, and ideology [10]. Only when pre-service teachers are able to rigorously build on their critical competencies would they and their future students be able to avoid the risk of continuing to interpret, produce, and spread digital texts without critiquing the social–political issues that are encoded in media texts (e.g., inequality, gender, racial and sexual oppression, religious discrimination, and political biases) [24].

Critical media literacies helped foreground the local micro-politics in literacy education that the teacher candidates have lived through. They enabled the researcher-teacher to engage in collaborative inquiry to problematize these micro-politics and envision alternatives for a universal or one-type-fits-all approach to literacy education [25].

## 4. Research Design and Participants

Methodologically, the project used digital ethnography to study pre-service teachers' online interactions on the social media platform of Teams [26]. Digital ethnography is a good fit for cross-border, online education projects, which is evident in two of the principal investigator's (PI) funded studies e.g., [27,28]. Digital ethnography helped the research team to explore communal practices through traditional and social media. To prepare literacy teachers for a larger-scale cross-border research project that connects youth from Hong Kong and Canada in a social networking space for digital story making, this study tried to cultivate a cross-border community of practice that could potentially foster global interconnectivity and collective exploration of equitable education. In this six-week project, the research team connected with four Hong Kong teacher candidates (See Table 1 for participants' profiles) in the online platform of Teams and used Google Docs to collectively pursue "real and relevant questions" [29] in Hong Kong and Canadian literacy education. Following ethics approval, an invitation to research was sent to teacher candidates at a Hong Kong university via the teacher education office. Interested teacher candidates signed the online consent. Since we invited the teacher candidates to be co-authors of the paper, we used pseudonyms for the participants in the blind review process.

**Table 1.** Participants' profiles.

| Participant Names | Major | Year of Study | Teaching Experience | Languages |
|---|---|---|---|---|
| Helen | Chinese language education (both Mandarin and Cantonese) | Year 2 | –Tutored Form 4 (equivalent to Grade 10) and Primary 1 (Grade 1) students at tutorial schools. Tutored students are local Hong Kong students and South Asian students who attended local Hong Kong schools. Taught subjects include Science, Chinese, English, and Math. –Volunteered to teach local Hong Kong students who could not afford the tutorial schools. | First language (L1): Mandarin; Second language (L2): English; Third language (L3): Cantonese |
| Angela | English education for primary and secondary schools | Year 4 | –Tutored Form 3 (Grade 9), Form 4 (Grade 10), and Form 5 (Grade 11) students. Students are local Hong Kong students who attended local mainstream schools. The taught subject is English. –Assisted in holding an English interest class for Form 1 and 2 (Grades 7 and 8) students. Participants were Hong Kong/Mainland Chinese students who attended a local mainstream school. | L1: Cantonese; L2: English; L3: Mandarin |
| Natalie | English language education and Chinese language studies | Year 4 | –Tutored Primary 6 (Grade 6) students in a local school (Subject: Chinese). –Provided online tutoring via Zoom to a group of primary students (including Grades 1-6) with SEN from HK local schools (Subjects: Chinese, English, Maths). –Tutored small groups of Form 2 (Grade 8) and Form 3 (Grade 9) in a local school(Subject: English). –Tutored Form 5 (Grade 12) and Form 6 (Grade 13) students in a local school. | L1s: Cantonese and Mandarin; L2: English; L3: Spanish |
| Jessica | English language education | Year 2 | Tutored Primary 3 and 4 (Grades 3 and 4) and Form 3 and 5 (Grades 9 and 11) students. | L1: Cantonese; L2: English; L2/L3: Mandarin |

In the 6-week project, pre-service teachers participated in (1) three rounds of collective digital instructional design; (2) online interactions on the social media platform of Teams and Google Docs about the designs; and (3) exit interviews about the impacts of the cross-border online project on their professional learning.

We deductively developed themes based on the selected theory and constructs of critical media literacies. We also remained open to emerging themes from the data. We analyzed the data by using a constant comparison method (CCM) e.g., [30]. To achieve a fit between data and categories/theories, we compared the newly collected instructional design and exit interview data with existing data from online interactions and selected

theories. We also adopted Handsfield's [31] modified CCM model and addressed the poststructuralist's critique of CCM as reducing complicated meanings and events into simplified aggregates. Using this modified model, we assigned a "third space" for data that fell into more than one category and for those that did not fall into any category at all but would illuminate teacher candidates' collective exploration and the impacts of the cross-border, online professional learning project. For instance, in analyzing the interview data, we found that most participants commented on the positive impacts of the project on their future pedagogical efforts to promote diversity and equity. However, Angela was the only one that expressed the concern that even though the component of promoting diversity is evident in the instructional design, she might not choose to do so in her real-life teaching practice. Though her comment did not fall into the theme, we highlighted it in the report to illuminate the potential challenges to enacting critical media literacies in certain educational contexts.

## 5. Findings

The presentation of the findings below centers around the major changes that teacher candidates made in the drafts of their collective instructional design. We present the related synchronous conversations on Teams and asynchronous feedback via Google Docs that informed these changes and teacher candidates' collective exploration of using critical media literacies to promote diversity and equity. We also draw on the exit interview data to show teacher candidates' perceived effects of the cross-border, online professional learning experience.

### 5.1. Identifying Real and Relevant Educational Issues

Earlier Teams conversations focused on ice-breaking and identifying the real and relevant issues for their professional learning about using critical media literacies to promote diversity and equity. The principal investigator (PI) and the teacher candidate participants shared their learning and teaching experiences and the challenges they encountered in online learning and teaching during the pandemic in Hong Kong and Canada.

All four teacher candidates communicated the challenges that they had encountered in online teaching during the pandemic. They also shared virtual tools and platforms that they found helpful to increase learner engagement in online learning. In the first Teams meeting with Helen, she shared that her younger Primary 1 (Grade 1) students could not pay attention to her teaching in the online classes and she felt "frustrated". Trying to find ways to better engage students online, she found a website where teachers could design interesting games based on their teaching content. She also added animations and interactive games in PowerPoints to increase her students' online engagement. Based on her tutoring experience, Jessica also found that it was difficult to motivate students to learn in an online environment. She recounted that it was almost impossible to know what her students were doing on the other side when they turned off their web cameras. She also shared the online platforms that she found useful to track students' online learning progress, such as Quizzes, Socrative, and Kahoot. Natalie echoed the challenges because she started online teaching via Zoom during the pandemic. She agreed that it was easier to check on learner engagement in face-to-face learning than in online learning when her students turned off the cameras. However, both Angela and Natalie commented that online learning during the pandemic pushed them to explore new teaching tools and pedagogical practices. For example, in Natalie's summer bridging class for Grades 6 and 7 students, she found that the social networking platform of Seesaw increased students' engagement in sharing their homework and providing feedback to their peers. Angela had her teaching practicum at a secondary school and had one online class during the pandemic. Through these teaching experiences, she realized that approaches to engage learners were more pivotal in virtual learning than in face-to-face teaching. She recalled, "My partner and I paid a lot more attention to how we can get everyone involved, how we can get students engaged. We turn out to use methods that I have never tried as a teacher in my life".

In the Week 2 Teams meeting, all teacher candidates shared their prior learning experience and how they would like to make changes in the test-oriented education that they had experienced. Helen communicated that when she was a primary and secondary student, her teachers expected students to "do the paper and do a lot of homework so that we can get higher marks". She said she completed a lot of exercises and past exam papers, and things went well until she was in Form 4 (Grade 10). Her marks suddenly dropped despite her efforts. Later, she found that interacting with her peers inspired more new ideas. This experience encouraged her to communicate with her students about what they know, their weaknesses, and even to learn from her students. In the same meeting, Jessica recalled that her interest in learning English started when she was very young, in kindergarten. She started learning phonics with a teacher from New Zealand through games and storybook reading. She applauded the fun and non-stressful learning back then. However, she felt that, now as a tutor, parents expected her to focus on exam preparation, especially in Hong Kong, which, according to her, is "a very exam-oriented" context. For the Primary 4 to 6 (Grades 4 to 6) students, they need good results to get into good secondary schools. And for the Forms 1-3 students (Grades 7 to 9), they need good results to choose electives and to be better prepared for the public exams. She recounted changes that she would like to make as a teacher:

> *I do try to incorporate authentic materials, which I think is a pretty important step*
> *of learning . . . . It's definitely more fun than what the current syllabus can provide . . . .*
> *They can engage in it to be active learners or some materials that they can genuinely find*
> *interest in, rather than being forced upon them.*

Natalie responded and said that the authentic learning resources in YouTube videos helped her "ace the exams . . . because of the exposure to so much information on YouTube. I feel like it's easier for me to understand the reading passages, because sometimes I am so happy when I opened the test paper, and I thought 'Oh, this topic I come across on YouTube'".

In sum, in the initial interactions, teacher candidates identified real-life challenges in virtual learning during the pandemic and emphasized how new media and technologies could enable literacy learners to engage in authentic materials and real-life social contexts. Nevertheless, in the initial conversations, there were limited in-depth conversations about issues of equity and diversity in literacy education.

### 5.2. Addressing the Issues through Critical Media Literacies

#### 5.2.1. Changes in Conversations about Equity and Diversity

After the synchronous interactions through the first Teams meeting, the teacher candidates' asynchronous conversations continued on Teams, and they started to brainstorm how new media could be used to respond to their identified challenges in literacy learning during and after the pandemic. On August 2, Angela shared her initial ideas via a Teams posting about the instructional design. The posting elicited peers' and the PI's interactions. They asked questions for clarification, provided further suggestions, and specified ways they each could contribute to the collaborative instructional design (e.g., via Google Docs and Teams synchronous and asynchronous interactions).

In our earlier synchronous and asynchronous conversations in Week 1, teacher candidates tended to converse about their experiences using technology or media as engaging tools, but there were scarce conversations about equity and diversity. In the Week 2 Teams meeting, Natalie briefly talked about her students with special education needs. However, the challenges in online teaching during the pandemic that she mentioned were not specifically about challenges in special education. Helen also shared that most of her South Asian students could understand, but could not speak, Cantonese. She also volunteered four hours per day to teach local Hong Kong students who could not afford the tutorial schools. Her students were diverse in their socio-economic status and linguistic and cultural backgrounds; however, the earlier conversations did not address pedagogical practices that were used to support these students. On August 4 in Week 3, the team met on Teams and

discussed these initial ideas. When asked about the rationales for these instructional ideas, Angela communicated,

> *I think it may be about motivation like the grouping, the peer support, the authenticity of text . . . . They are all working together to make learning more interesting . . . .so that students are more willing to learn by themselves and not relying on teachers spoon-feeding them.*

The ensuing synchronous conversations show teacher candidates' focus on motivation as a psychological construct to engage students. The Week 3 conversations rarely addressed sociolinguistic issues about imbalances of power in relationships within literacy classes.

On August 6 (Week 4), Angela took the initiative and created the first draft of the instructional design. On August 11, the team had a synchronous Teams meeting to discuss this initial design. Helen started the conversation by expressing her preference for the debate activity. She said she would use debates for her Chinese language teaching classes because "in our Hong Kong Diploma of Secondary Education (DSE: university entrance examination for Grade 12 students in Hong Kong), we will have oral examination. So if we use debates in our class, that will help them to improve their communication skills". Later on, when discussing students' varied language competencies and needs, Helen communicated that "If my class has more students who need to improve their communication skills, I will use debate as a class activity; however, if some students in our class are not good at these aspects, maybe I will consider using another activity for them". Helen's feedback prompted the team to think about whether it is right to marginalize students who are not good at debating, especially in a language in which they are still developing competency. Natalie then proposed an idea about differentiated instruction to cater to students' varied competencies and strengths. She suggested in the Teams meeting:

> *If a kid is not really good at presenting or reacting quickly, we may arrange him or her as the first speaker because for a first speaker they only need to read the script. They can just have a really well-prepared script and read according to that . . . . Maybe the rest of the class can help them as the support group, like their intelligence bank or intelligent teammates who help them research if they prefer not to speak in front of people.*

Natalie was inclined to allow students to present or debate when "they feel like they are ready". But later, all the participants agreed with Angela when she proposed her idea of pushing students a bit. Angela said that, in Hong Kong, students are generally "rather shy", and creating more opportunities for debating can help them "gain confidence and encouragement to speak".

In a nutshell, teacher candidates' synchronous and asynchronous interactions started to show their focus shift from test-oriented practices to meaningful learning opportunities that cater to diverse student interests and competencies.

### 5.2.2. Problematizing Stereotypes through Critical Media Literacies

The teacher candidates' Google Docs interaction relating to Draft One posted in Week 4 started to show their initial awareness of incorporating critical media literacies. In the pre-module preparation, students were expected to watch YouTube videos about stereotypes and connect to their real-life experience so that they "become aware of past biases/prejudices/stereotypes" embedded in the media and "recognize how limited perspectives/knowledge leads to biased perceptions" (direct quotes from Draft One). In the August 11 Teams meeting, Natalie asked whether she could add YouTube videos of authentic settings. Angela responded,

> *If we want to get students to do more analyses, we may have to change the short videos because the original videos I am thinking about may be too explicit [about what people think about Americans or Chinese]. If you want to incorporate critical media literacies in that part, then maybe we have to choose videos that have subtly embedded stereotypes.*

After the August 11 meeting, Angela made changes to Draft One and changed the topics for the debate (See Table 2). Responding to Natalie's inquiry about the key features of critical media literacies, the PI posted pertinent information on Teams on August 11. On August 17, Natalie added the new component of critical media literacies to the instructional design.

**Table 2.** Changes to the stereotypes activity.

| Draft One | Draft Two |
| --- | --- |
| Students watch videos about how Chinese and foreigners view each other and answer the question: Why do you think they think of each other in this way? | Teacher candidates agreed to add three YouTube videos about how Chinese/Japanese/Korean people feel about stereotypes of themselves. They also added an introduction to critical media literacies to inform students that "a critical attitude should be held towards information received, especially online information". They also prepared questions for students to brainstorm about the videos: Who created these posts? What creative techniques are used to attract my attention? What lifestyles, values, and points of view are represented in, or omitted from, this message? What was the purpose of these posts? |

Regarding the changes in Draft Two, Natalie posted on Teams,

> *[2021-08-17 1:59 p.m.] Natalie: . . . . I have added an activity about introducing critical media literacies based on your explanation to the part of YouTube videos about stereotypes. In the activity we will ask students to distinguish which one is a more authentic/trustworthy source by checking its source of information and different opinion[s] being presented in the source, thus leading students to think about holding a critical attitude towards different media.*

Natalie also pondered pedagogical approaches to critical media literacies. As she asked in Google Docs, "Would the questions be useful for promoting critical media literacies? Or simply lecturing the principle of critical media literacies during lesson would be better?"

Keeping brainstorming and revising the instructional design, the teacher candidates made salient additions about critical media literacies to Draft Two.

### 5.2.3. Realizing the Ideological Model of Literacy Education

The online conversations about the debate topic in the instructional design continued both synchronously and asynchronously. The conversations reflected the teacher candidates' changed perceptions of literacy, which concerned reproducing power in certain languages and cultures. The ensuing changes in the instructional design are reflected in Table 3.

**Table 3.** Changes to the debate topic.

| Draft One | Draft Two |
| --- | --- |
| Debate Activity: In groups, choose one culture and argue on behalf of it being the worthiest to retain. | Debate Activity: In groups, think about one way that would be the most useful for the preservation of customs (the groups can seek examples of customs in the materials). |

After Angela posted Draft One on Google Docs on August 6, Jessica raised a question about students' feelings if their cultures are deemed by their peers as unworthy of retaining. The question elicited their peers' responses within Google Docs:

> *Jessica: I'm a bit worried that this may upset a few students if their culture loses—is it possible that we can change it to which they should pick "which is the most interesting"?*

> *Angela: I think we may also avoid customs that belong to students' own cultures in order to avoid offending students from a certain cultural background in the debate. We can emphasize that there is no absolute winner in the debate, since every culture has its own value . . . .*

> *Natalie: Would it be possible if we restrict them to choose their opponents' cultures? That way they can learn to understand each other's cultural background and appreciate them when they are debating for the culture.*

In the August 11 Teams meeting, the conversation about "worthiness" and "unworthiness" continued. The PI cited Ostler's [32] paper that by the year 2100, the human race will have lost about 50% of the languages that are alive today, which means that every 14 days, a language dies. Then Natalie commented that when a language dies, the related cultural heritages and histories also die. Later, the team explored synchronously on Teams about how the proposed activity in Draft One should be about helping preserve languages and related cultural customs:

> *Angela: So culture becomes perspectives or aspects, but not necessarily the understanding for different preservation methods, but they are not choosing a culture. The culture is simply an example.*

> *Natalie: So maybe the focus will be on which method is a better one. Maybe using culture as an example as Angela just mentioned . . . . I remember Jessica said that maybe some students will be sad if their culture lost in the debate. And I was thinking whether we can restrict our students to choose each other's culture . . . .*

> *Helen: I think we can think about how to make some special topics for them, and not about comparing which culture is better, but about promoting some culture. For example, we can promote our culture through television or through movie to let others see what a beautiful culture we have . . . . We can let them brainstorm how they can promote their cultures. That could help them have a better understanding about their culture.*

> *Jessica: . . . . Will it be possible for students to interview each other as well? Which can be a good class activity for them to use English in the process maybe. Actually I am not really sure about the argument about which one is the better culture. That could be leading to cautions. Like Helen mentioned, maybe we can also brainstorm how to preserve the culture instead.*

The continued conversations in Teams below started to show teacher candidates' awareness of the interconnectedness of languages and cultures and power imbalances associated with dominant and marginalized languages:

> *Natalie: I think that even though one language may not be the fittest one among all the languages, we still need to preserve it because when we are saying that it is the fittest in terms of whether it is valuable for daily usage or whether quite a number of people are using it as the official language. . . . . A language itself helps us understand other languages, they are all related . . . . It is not the fittest at all, but it is still valuable because it helps us understand what it was like back in the days, maybe also helps us to investigate culture and also the social background, or a specific period of time that it is under foreign influence . . . . We cannot judge it by simply saying that it is not popular so we should not preserve it.*

> *Angela: Actually I really agree with Natalie. She considers language as artifacts, the reflection of social, historical, economic, political, cultural things . . . . But for the Canadian linguistic contexts, on a macro perspective, I really agree that we should preserve languages which are not as popular as official languages. But in a micro perspective, if I were a Canadian born Asian, I would agree that I have to learn English first because I live in this environment . . . . I am not saying that we should totally abandon Cantonese,*

> *but I may spend more time in English first until I have mastered that or I can have extra efforts that I can spend on the mother tongue of my parents.*

Teacher candidates' views about maintaining minoritized languages or promoting dominant languages still diverged, but they started to be aware of the ideological model of literacy and agreed on the change in the debate topic to celebrate diverse students' cultural customs. Their transformed perceptions of literacy showed their awareness of coercive power relations between dominant and marginalized languages and the impacts of marginalization and isolation that might be incurred by certain pedagogical practices or class activities. The changes in the debate topic also reflect their transformed practice to encourage power negotiations in literacy classes.

Draft One shows the teacher candidates' salient awareness of giving students a sense of ownership over learning. Jessica admitted in the Week 2 Teams meeting that she was from a "very privileged" family, and that was why she was "immersed in English environment" since her early childhood education. Things came naturally to her with respect to her success in English learning, and now, as a tutor, she found it "very hard to motivate students". Jessica was aware that she needed to motivate her students with "practical needs", but she still needed to make English learning meaningful for her students, who might not see the purpose of learning English in the Hong Kong context, in which English is one of the official languages but not the most commonly used language. In our asynchronous conversations in Google Docs about the rationale of incorporating purposeful learning, Angela responded, as she initiated Draft One,

> *In Hong Kong, I think students often feel they are learning for exams or simply because the curriculum says so. Therefore, I am trying to give them a sense of purpose and tell them that they are learning for themselves, which I think would subtly give them a bigger sense of ownership over their learning.*

Looking at the component of teaching vocabulary and grammar and sentence structure as the target knowledge in Draft One, Jessica suggested via Google Docs that "We can think about how we can incorporate this into the design without making the task out of place—maybe structures that can be commonly found in brochures/tourism websites, so students can relate it back to what they have seen". Jessica asked in Google Docs whether her peers would consider using "different quiz platforms to make this more exciting and less heavy for the students". To make the Q&A session "interactive", Natalie also agreed, "I think using multimedia can make the Q&A less quiz-like. Maybe we can use online quiz platforms like Kahoot/Quizizz to engage them and perform the quiz as a pop-quiz game". On August 17, Natalie inserted an interactive quiz in Draft Two (See Table 4).

**Table 4.** Addition of interactive online assessment.

| Drafts One and Two | Final Design after Draft Two |
|---|---|
| Knowledge Input: Guide students through authentic materials (materials not limited to written texts; can be audio, video, etc.) about different customs<br>Source: YouTube, official tourism websites (subject to possible modification to lower difficulty)<br>Target knowledge:<br>Vocabulary<br>Grammar/sentence structure<br>Genre writing skills → how to write informative articles? | On August 17, Natalie added a few questions to engage students in an interactive assessment. The questions are about cultural practices, their popularity, and some interesting facts about different cultures.<br>1.What is one of the five most spoken languages in America? (popularity of a language)<br>2. It is usual for a Spaniard to have lunch at 2 p.m. and dinner at around 8–9 p.m. (true/false) (cultural practice)<br>3. What is the most popular language in the world? (popularity of a language)<br>4. There are approximately 1500–2000 spoken African languages. (true or false) (diversity of cultures/languages) |

For Draft One, the teacher candidates expressed that their rationale for using authentic materials was to motivate students to learn online sources on their own. However, the final design shows their interest in making learning interactive and enjoyable for literacy learners.

*5.3. Teachers' Perceived Effects of Cross-Border, Online Professional Learning*

5.3.1. Changed Perceptions of Using Critical Media Literacies to Promote Diversity and Equity

In the exit interviews, teacher candidates expressed their changed perceptions about how critical media literacies could be used to promote diversity and equity. Helen commented on how new media such as YouTube could be used in instructional designs to cater to diverse students' needs and linguistic repertoires. Helen expressed that through this project, she felt that she had increased awareness of promoting educational equity for diverse students. She shared her thinking about her future practices:

> *I should provide more diverse resources to students. Or add some critical thinking, such as writing more essays, to enhance their creativity and inspire them to think. I will also take care of the needs of non-Hong Kong students, such as Filipino and Indonesian students.*

Trained in Chinese teaching, Helen said she would use "bilingual pedagogy" in her future teaching when teaching South Asian students in Hong Kong. Instead of focusing on these students' competencies in Chinese, she would focus on their interests and their linguistic strengths in English. Natalie's and Jessica's views concern Hong Kong students' diverse heritage languages. Natalie thought the teaching method prevailing in local Hong Kong schools "is not really catering for students who have different mother tongues". She also had concerns that in most of the local schools in Hong Kong, "there's not much support for those students", and schools "don't really have enough resources to, like particularly design a tailor-made textbook or a tailor-made schedule for those students". Natalie also concurred that incorporating different media to problematize "stereotypes towards different groups of people" in their instructional design helped enhance her awareness of promoting equity, diversity, and inclusion in instructional designs. She said that in the project, the teacher candidates tried to create an instructional design that is "inclusive about different cultures and customs around the world". Jessica went further and talked about how the project influenced her understanding about how instructional designs for literacy classes could incorporate new media technologies to promote equity and cultural diversity. She admitted that she gained insights into how to design literacy activities with digital tools to encourage learners to "share their own cultures, discuss their values openly . . . and develop respect for each other". Jessica further explicated that using interactive virtual platforms and encouraging students' self-exploration on the internet of different cultures would enable the authentic purposes of learning and enhance students' cultural understandings of their peers. Angela also agreed that the project made her think about promoting equity through new media use; nevertheless, she also worried that the equity-oriented module that they designed would not fit the test-oriented school culture in Hong Kong. Angela mentioned three reasons why she could not promote educational equity and diversity in her future teaching. Firstly, "it could more demanding if I pay a lot of attention to equity". Secondly, teaching in Hong Kong requires more effectiveness and they "do past papers, do past papers, and do past papers; seldom creativity gets incorporated in senior secondary schools". Thirdly, "cultural diversity is not really explicit in normal classrooms". Though the instructional design portrayed students as from different backgrounds, in her view, most of the schools in Hong Kong were "mainly homogeneous". She said that non-local students were "mainly from mainland China" and that most of the ethnic minority students did not study together with local students but in designated schools.

Natalie was the only participant who emphasized how the process of brainstorming ideas to promote equity and diversity enhanced her awareness of providing equitable opportunities for students with special education needs (SEN). She said,

> *I think we also included different media or some interactive tools like Kahoot in our instructional design . . . . I think maybe using those kinds of tools would be easier, for students especially those who have SEN to keep a longer attentional span because we hold their attention by providing different media for stimulation, and including different teaching methods in our instructional design for interaction, or different tools to evaluate students.*

In sum, all four teacher candidates concurred that the project helped enhance their awareness of promoting diversity and equity through the use of critical media literacies.

5.3.2. Transformed Perceptions of Literacy

In the interview conversations, three teacher candidates expressed their changed perceptions of literacy through this project.

Angela agreed that before the project, "I was only concerned about transmitting knowledge of language". She said, "Normally I would pay attention to how I can make my students understand different concepts in English language. But now I am more aware of the importance and the results of cultural and value education embedded within English education". Helen communicated about how her prior learning experience influenced her teaching. She said that her teachers would focus on high marks in their teaching, which "influenced my earlier teaching when I emphasized on helping students getting high scores". She said, later, she realized that exploring with students about their curiosity was very important: "If teaching attends to their curiosity, they will be more motivated to learn the articles and learn Chinese and other subjects well". She said that after the project, she would think about "how to arouse students' interests in the Chinese articles that they learn and how to make the learning meaningful to motivate them to learn". Jessica mentioned her changed perceptions of literacy from being more skills-focused to equity-oriented. She recounted,

> *Definitely it has broadened my perception on what literacy is . . . . Talking about literacy education, my thoughts were forefront to empower students by teaching them English skills . . . .But I haven't given so much thought on how the content or the interactions of students within the lesson can also impact the students in a way . . . .to promote equity.*

Jessica agreed that opportunities for literacy learners to experiment with different media, such as visual, audio, or digital, would give students a sense that the English lessons were "not just about English skills or other thematic knowledge". Jessica recounted that in one of the Teams meetings, we discussed students who did not speak the dominant languages of schooling. She said that the conversation "really prompted me to think about situations like this or whether there are any creative ways to solve this problem (pedagogically). We definitely can't use our conventional ways to just teach sentence structures or vocabulary".

Two teacher candidates expressed their interest in exploring diverse conceptual orientations of literacy education because of this cross-border professional learning project. Helen would like to join similar cross-border projects in the future to learn more about different ways or theories of literacy education. She shared in the interview, "I feel that I can learn a lot from teaching practices outside of Hong Kong". Natalie also agreed that the major impact of the project on her was knowing about the educational crisis in a bigger context and broadening her horizons about what perspectives of literacy education were taught in Hong Kong and Canada. Natalie mentioned that the exam-oriented culture in Hong Kong constrained teachers' willingness to "change our teaching methods", but connecting with pre- or in-service teachers around the world would "help us understand how education theories are implemented in different cultural and social contexts". To cite her, "I think it is necessary just for teachers to broaden their horizon and to know about theories and different perspectives of education". Natalie agreed that the professional training project helped her make connections between her prior teaching and

learning experiences and various literacy theories and made her think about how she could "put the theories into practice under a real local context".

All of the teacher candidates emphasized that it was important for them to be aware of the manipulative power of media, such as challenging the stereotypes of diverse groups of people embedded in media. For example, Natalie shared that "I think it is really important for students to also learn about how to distinguish which resource is more trustworthy or authentic given that we have different kinds of fake news online". Natalie reflected on their addition to the instructional design to encourage learners to make videos collectively (See Table 5).

**Table 5.** Addition of learners as new media designers.

| Drafts One and Two | Final Design after Draft Two |
| --- | --- |
| Group project: (also evaluation)<br>• Instruction: Ask students to collect information about one foreign custom that they think their classmates have never heard of and that would be the most interesting<br>• Present their result in front of the class and explain why they chose this custom | Group project: (also evaluation)<br>• Instruction: Ask students to choose one custom they would like to help preserve and use the methods mentioned in the previous debate to promote it after researching<br>• For example, they can do a video about the history of lion dance and post it on YouTube so that more people learn about it<br>• Present their methods and products in front of the class |

She commented on how supporting literacy learners as media designers could transform learners from being passive media consumers to active learners. She shared,

> Yes, I think maybe instead of playing videos in class, another way for them to be more engaging would be asking them to work in a group, to shoot a video about their own culture before we actually write the instructional design . . . . I think this would be collaborative because they will be able to work in groups with students that are from different cultural backgrounds, or have different knowledges of different customs . . . . So I think maybe being the creators of the videos will be more interactive and also will be more engaging than playing videos in the class.

However, as is evident in the interview conversation, the purpose of incorporating media design stopped at motivating students instead of supporting them to be active participants in social change.

### 5.3.3. Teachers' Perceived Limitations of the Project

In the interviews, teacher candidates also shared their views about the limitations of the project.

Two teacher candidates wished that the professional learning projects would have used more of their knowledge of technology use and bilingualism. Helen shared that she "really improved English communication and writing skills" through the project. But because of the length of the project and the scope of the collaborative instructional design, Helen did not think her knowledge about how to use new media in literacy education was well harnessed in the project. She said that, besides the use of YouTube and Kahoot, the instructional design did not incorporate the games that she had found helpful to support students with varied abilities and interests. Jessica also commented that most conversations were conducted in English, which might have limited teacher candidates' use of their multilingual repertoires.

Participants also communicated about the constraints of the Teams platform in cross-border professional training projects. Jessica said, "we had to change to being a guest of the PI's university in Teams" and that they did not often use Teams in Hong Kong. She also shared that asynchronous conversations on Teams "appeared to be more serious and

it's less personal . . . . That would be less enjoyable for me". Natalie expressed that she enjoyed the synchronous conversations and brainstorming more than the asynchronous activities. But she also agreed that asynchronous interactions were helpful when it came to refining the instructional design. The major challenge for her in cross-border professional learning was her personality, as she was afraid of "speaking things wrong". She suggested that more time would allow for more thoughts to emerge for the instructional design. Angela enjoyed the synchronous conversations. As she shared, "Teams is nice because it is multifunctional, like it allows real-time meeting and it allows synchronous discussion". Angela took the lead in the instructional design and commented that the exchange of ideas pushed her "boundary to incorporate more cultural value, equity, and diversity issues". However, she wished to see more collaboration in the process of instructional design. Natalie felt that, starting as strangers, the exchange of ideas through Teams and Google Docs was "rewarding because every single time we found a better way to design our instructional design".

In sum, interview data show teachers' increased awareness of using critical media literacies to promote diversity and equity. The findings also relate their perceived changes in terms of conceptualizing literacy.

## 6. Conclusions, Discussion, and Implications of Study

The research found teacher candidates' transformed perceptions and practices in the following aspects: (1) their shifted focus from training their students' decontextualized language skills to nurturing their critical media skills; and (2) the shift from a deficit-oriented view about what literacy learners could not do to an asset-oriented view about the linguistic and cultural repertoires that those students could bring to diverse classrooms.

The cross-border professional learning facilitated teacher candidates' collective knowledge creation and inspired their ethical insights about how critical media literacies could affect educational equity locally and globally. First, teacher candidates demonstrated their changed perceptions of literacy education, from focusing on decontextualized skills residing in individuals to addressing the cultural and ideological assumptions of literacy education [33]. The behavioral and cognitive paradigm of literacy sees imparting knowledge through "linear and staged curriculum" as crucial for all literacy learners [34], p. 6. Literacy learners' own interests and creativity (i.e., affective dimensions) are to a great extent neglected in literacy curriculum and instruction that focus on decontextualized skills and knowledge. Throughout the project, most teacher candidates explicitly expressed the importance of authentic media resources and meaningful learning to engage students. It is important to note that the teacher candidates' position was mainly about using socioculturally constructed symbolic tools to facilitate learners' cognitive development. Such a positioning primarily deals with the influences of social processes upon individuals' psychological construction of meaning. Though teacher candidates gradually showed enhanced awareness of power relations embedded in diverse languages and within literacy classrooms, there is still space in the project to explore the social and ideological aspects of literacy education; for example, how various forms of literacy are socioculturally, politically, and ideologically situated phenomena e.g., [33]. Second, teacher candidates showed their shifted focus from test-oriented teaching to asset-oriented teaching. Test-oriented education produces passive learners who are "less able to deal effectively with a highly complex, interdisciplinary, intercultural, mediated social world" [14], p. 239. Scholars also problematized the fetishization of standardized literacy testing and prescriptive curriculum that might hinder meaningful, improvisational, and heterogeneous meaning-making e.g., [35]. The teacher candidates drew on their previous learning and teaching experience and intended to incorporate meaningful learning opportunities and media creation activities to energize literacy learners. As Share [13] argued, "It is often through the process of making media that students deepen their critical understandings and develop a sense of empowerment by producing counternarratives or telling stories rarely heard" (p. 23). Such a "wealth model" [36] is in contrast to the deficit approaches and foregrounds diverse literacy learners' distinctive

funds of knowledge (e.g., skills, knowledge, and cultural resources and heritages) [37]. Ensuring equity among learners during COVID-19 shall be an important element in the new normal of education (e.g., [38]). Equity can be taught and practiced in online learning as it constitutes a crucial element of social justice that is much needed during troubled times [39]. It is necessary for future studies to involve school leaders, teachers, parents, and students and develop their capabilities to critically analyze the ethics of online learning and advocate for social justice and equity in alternative ways. Angela, one of our teacher candidates, rightly pointed out the real-life challenges for literacy teachers in incorporating pedagogical elements to promote equity and diversity in local Hong Kong schools. Future research and teacher education could further such collective exploration to enhance pre- and in-service teachers' confidence in teaching "concepts of social justice and equity in ways that make that learning powerful and integrated into minds, hearts, and bodies" [39].

Changes in the teacher candidates' collective instructional design reflect their changed perceptions of literacy as ideological and encoded with the power imbalances of dominant and minoritized languages and cultures. Those changes manifest teacher candidates' explicit exploration of equitable learning opportunities that promote diverse learners' heritage languages and cultures, such as the problematization of stereotypes in YouTube videos and the collective video making activity of preserving heritage cultures (e.g., YouTube video making about the history of lion dance). As Kellner and Share [22] noted, "When groups often under-represented or misrepresented in the media become investigators of their representations and creators of their own meanings the learning process becomes an empowering expression of voice and democratic transformation" (p. 372). In this study, the component of supporting literacy learners as creative media designers is not as explicit as the element of encouraging them to be critical media users. This refers to a way forward for future teacher education research on critical media literacies to nurture agentive participatory citizens. As Janks [40] contended, "deconstruction without reconstruction or design reduces human agency" (p. 178).

Virtual, cross-border professional learning also offered teacher candidates the opportunity to experience the constraints of technological platforms and tools, which would deepen their understanding of how to use media and technologies to promote equity and diversity in online teaching. In the exit interviews, teacher candidates shared their reflections on the limitations of Teams as a social network platform. Examining anti-oppressive pedagogies in online learning, Migueliz Valcarlos et al. [41] maintained that "technology platforms and tools that educators use to host their classes embody social, cultural, and political values and biases" (p. 355). Future teacher education research could further the examination of the associated biases that "enable and constrain conversations, relationships, and learning in different ways" in virtual learning platforms (p. 356). Involving teacher candidates in online professional learning projects could help them to be more reflective of the different critical situations that their learners may be experiencing in virtual learning spaces.

The pandemic has brought to the fore inequalities among various student populations, and emerging research suggests that the design of remote learning can perpetuate or ameliorate these inequities [42]. This teacher education study considered the negative impacts of the COVID-19 pandemic upon education systems worldwide and redressed educational inequities through collective problem-solving between people across cultural and linguistic backgrounds. The study highlights the importance of exploring asset- and equity-oriented literacy education that could enhance pre-service teachers' transnational links to ensure equitable learning opportunities and to support education in unpredictable crises.

**Author Contributions:** Conceptualization, Z.Z. and Q.M.; methodology, Z.Z. and I.L.; validation, Z.Z. and I.L.; formal analysis, Z.Z.; investigation, all co-authors; writing—original draft preparation, Z.Z. and Q.M.; writing—review and editing, Z.Z. and I.L.; supervision, Z.Z. and I.L.; project administration, Z.Z. and I.L.; funding acquisition, Z.Z. All authors have read and agreed to the published version of the manuscript.

**Funding:** This research was funded by Western Strategic Support for SSHRC success; grant number [R5371A13].

**Institutional Review Board Statement:** The study was approved by the Research Ethics Board of Western University (Project ID: 118826 and date of approval: 2 June 2021).

**Informed Consent Statement:** Informed consent was obtained from all subjects involved in the study.

**Data Availability Statement:** Data is unavailable due to ethical restrictions.

**Conflicts of Interest:** The authors declare no conflict of interest.

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
