# Peer review of "Cross-Border Dialogues: A Collaborative Instructional Design Inquiry to Promote Equity and Diversity"

_education, doi:10.3390/educsci13060567_

Round 1

Reviewer 1 Report

An interesting research focus that identified the learning explored by pre-service teachers. Maybe more approaches should be like this with pre-service professionals? The learning journeys are articulated thoroughly as the pre-service teachers navigate their way through teaching approaches, methods and delivery frameworks before actual practice settings. The exploration of a much more student led asset approach, inclusive techniques and teacher adaptability was good to see. Consideration of embedding socratic questioning and critical analysis techniques offered possibilities for pre-service teachers to create meaningful pedagogical experiences. Linking this to the pandemic context reaffirmed that teaching and learning was affected globally with similar issues to contend with, even within different cultural contexts.          

Suggestions:

- the findings section (5) is quite long. Is all the narrative background needed or could this be reduced offering shorter summaries?

- assuming the named participants (pre-service teachers) names a pseudonyms? Unless they given ethical approval to use their names? 

Just a few grammatical errors.

Author Response

Dear Reviewer and Editors,

We really appreciate the review feedback. It has been very helpful in the process of polishing the paper for publication in Education Sciences. We’ve used track changes for the revision. We’ve kept tracked changes for deleted texts for Section 5 Findings based on the reviewer’s feedback about making this section less lengthy.

Should you have questions, please don’t hesitate to let us know.

Best regards,

Co-authors

  1. The findings section (5) is quite long. Is all the narrative background needed or could this be reduced offering shorter summaries?

Response: We’ve tried to simplify texts wherever possible. Please see the tracked, deleted texts.  

  1. Assuming the named participants (pre-service teachers) names a pseudonyms? Unless they given ethical approval to use their names? 

Response: To empower pre-service teacher participants, we included them as co-authors of the paper. Therefore, we will use real names in the published paper, though we are using pseudonyms for blind review. Taking the reviewer suggestion, in the section of “Research Design and Participants”, we’ve added: “Since we invited the teacher candidates to be co-authors of the paper, we used pseudonyms for the participants in the blind review process.”

  1. Just a few grammatical errors.

Response: We’ve done a thorough proof-reading and made corrections throughout the paper.

Reviewer 2 Report

The manuscript has an empirical character and it is focused on the cross-border teacher education. There was realized project, which could improve the level of education in some cases. Authors in the examination part of the manuscript used interview as the method. The text is written on high level without any typographical errors. All used references are cited in correct form in the text and even in the References. Tables included correct kinds of information. I have got only two comments, which could be easy incorporated in the text, they are presented below.

1. The main aim of the study is unclear. Please add them at the end of the theoretical background of the manuscript and also add it into the abstract of the manuscript.

2. The second comments is regarding to methodology. Please add some sentences (1 – 2) about it, how the responses of respondents were analyzed.

And also please add only one sentence about it, if the names of respondents are real or they are acronyms.

I hope my comments are helpful

Author Response

Dear Reviewer and Editors,

We really appreciate the review feedback. It has been very helpful in the process of polishing the paper for publication in Education Sciences. We’ve used track changes for the revision. Most deleted texts were not tracked to ensure a cleaner copy for the 2nd review. We’ve only kept tracked changes for deleted texts for Section 5 Findings based on the reviewer’s feedback about making this section less lengthy.

Should you have questions, please don’t hesitate to let us know.

Best regards,

Co-authors

  1. The main aim of the study is unclear. Please add them at the end of the theoretical background of the manuscript and also add it into the abstract of the manuscript.

Response: I’ve added the purpose statement in both Abstract and Section One of “Context”: “This teacher education project intended to examine how academics and pre-service teachers in different geographic locales could collaboratively explore equitable learning opportunities for diverse learners by using critical media literacies to respond to interconnected global crises”.

  1. The second comments is regarding to methodology. Please add some sentences (1 – 2) about it, how the responses of respondents were analyzed.

Response: I’ve made the methodological approach explicit at the beginning of “Research Design and Participants”. Please see the added and revised content: “Methodologically, the project used digital ethnography to study preservice teachers’ online interactions on the social media platform of Teams (e.g., Hjorth et al. 2017). …. Digital ethnography helped the research team to explore communal practices through traditional and social media.”

  1. And also please add only one sentence about it, if the names of respondents are real or they are acronyms.

Response: To empower pre-service teacher participants, we included them as co-authors of the paper. Therefore, we will use real names in the published paper, though we are using pseudonyms for blind review. Taking the reviewer suggestion, in the section of “Research Design and Participants”, we’ve added: “Since we invited the teacher candidates to be co-authors of the paper, we used pseudonyms for the participants in the blind review process.”

Round 2

Reviewer 1 Report

Reviews met.